# The Effect of BMI on Blood Lipids and Dyslipidemia in Lactating Women

**DOI:** 10.3390/nu14235174

**Published:** 2022-12-05

**Authors:** Lianlong Yu, Xiaohui Xu, Wei Yu, Liyong Chen, Shixiu Zhang, Yanmo Li, Meina Tian, Changqing Liu, Xiaoyan Luo, Yiya Liu

**Affiliations:** 1Shandong Center for Disease Control and Prevention, Jinan 250014, China; 2Yantai Economic & Technological Development Area Center for Disease Control and Prevention, Yantai 264006, China; 3Department of Clinical Nutrition, Qilu Hospital of Shandong University, Jinan 250012, China; 4Department of Nutrition and Food Hygiene, School of Public Health, Cheeloo College of Medicine, Shandong University, Jinan 250012, China; 5Hebei Center for Disease Control and Prevention, Shijiazhuang 050021, China; 6Guizhou Center for Disease Control and Prevention, Guiyang 550004, China

**Keywords:** dyslipidemia, BMI, lactating women, blood lipids

## Abstract

Background: This study aimed to explore the correlation between body mass index (BMI) and dyslipidemia and the optimal cut-off point for BMI to distinguish the risk of dyslipidemia in lactating women. Methods: A total of 2295 lactating women subjects were included in this study, all within 2 years postpartum. All samples were from “China Children and Lactating Mothers Nutritional Health Surveillance (2016–2017)”. BMI, blood lipids, demographic information, lifestyle habits, and other serum indicators were obtained in this survey. Generalized linear model, logistic regression, restricted cubic spline (RCS) and ROC curve analysis were used to evaluate the relationship among BMI, blood lipids, and dyslipidemia. Results: BMI in lactating women was positively correlated with total cholesterol (TC), triglyceride (TG), low-density lipoprotein cholesterol (LDL-C), but negatively correlated with high-density lipoproteincholesterol (HDL-C) (*p* < 0.05). Higher BMI in lactating women was associated with higher ORs of dyslipidemia (hypercholesterolemia, hypertriglyceridemia, high-LDL-cholesterolemia, low HDL-cholesterolemia) (*p* < 0.05). These associations were stable across age groups, breastfeeding child age (months), parity, physical activity level, fasting plasma glucose (FPG), and hemoglobin. These factors did not interact with this relationship (*p* > 0.05). The optimal cut-off point for BMI was 24.85 kg/m^2^ determined by using ROC analysis, which can distinguish the risk of dyslipidemia. Conclusions: BMI was positively correlated with risk of dyslipidemia. Maintaining an ideal weight may prevent dyslipidemia in lactating women, and BMI is recommended to be controlled below 24.85 kg/m^2^.

## 1. Introduction

Body mass index (BMI) is a common measure of obesity in clinical and public health practice. Obesity is intimately associated with dyslipidemia [1,2] accompanied by a series of diseases, including cardiovascular disease, diabetes, and metabolic syndrome [3,4]. Usually, BMI is widely used to assess whether someone is overweight or obese, because it is convenient, non-invasive, and easy to obtain [5]. High BMI is an important risk factor for poor global health outcomes and increased burden of health expenditures [6,7,8], causing about 5 million deaths globally in 2019 [9]. Increased BMI is directly related to dyslipidemia according to previous literature based on the US adult population [10]. Likewise, according to a prospective cohort study of 5195 adults from the United States, Finland, and Australia, the frequency of dyslipidemia gradually increases with increasing BMI (*p* < 0.0001) [11]. The prevalence of dyslipidemia is a major global public health problem. Meanwhile, dyslipidemia is as associated with increased risk of insulin resistance and cardiovascular disease [12]. The Framingham Heart Study showed that maternal dyslipidemia before pregnancy was closely related to their own health and the future health of their offspring [13]. Although there have been many studies on women’s health previously, limited research has focused on the health of women during their special period of breastfeeding. 

Lactation is an important period of metabolic tissue development and reorganization [14]. Simultaneously, overweight and obesity have been widespread in lactating women [15,16], which could induce or develop to their midlife obesity [17,18]. Several studies have shown that excess weight gain in the process of lactating increases risk of metabolic syndrome [19,20]. Different BMI and blood lipids may potentially impact on breast milk composition and the process of lactation itself [15]. There are clear studies showing that maternal blood lipids and BMI can influence cardiometabolic disease in the mother and her children [12]. Therefore, further understanding around the health problems of nursing women is required. Based on the above literature, lactation, as a special physiological period, has different metabolic characteristics from other adult periods. It is one of the factors affecting the relationship between BMI and dyslipidemia.

Until now, little has been known about the relationship between BMI and lipid metabolism among lactating women especially in the Chinese population. Though most postpartum women focus on maintaining healthy weight after delivery [21], there is no consensus on which range of BMI is the minimal risk for dyslipidemia. In this regard, the previous literature has not fully and efficiently used BMI as a convenient and potentially relevant index. Studying the relationship between BMI and dyslipidemia among lactating women is highly significant to the health of women. Thus, our study explored the correlation and the optimal cut-off point for BMI to distinguish the risk of dyslipidemia in lactating women. 

## 2. Materials and Methods

### 2.1. Study Population and Data Collection

A total of 2295 lactating women subjects were included in this study, all within 2 years postpartum. This study was based on the “China Children and Lactating Mothers Nutritional Health Surveillance (2016–2017)” survey approved by the Chinese Center for Disease Control and Prevention (China CDC). This survey was organized by the China CDC, which organized nutrition experts to conduct argumentation, design questionnaires, and select survey sites, and was carried out by the provincial, municipal, and county CDC. Since lactating women were selected as the research sub-subjects in this survey, we chose this project as our data source. China CDC staff randomly selected the districts according to demographic proportion and economic distribution. All the staff involved in the intended sample survey have received unified training and assessment from China CDC to ensure that they work in accordance with unified standards. Hebei, Guizhou, and Shandong provinces, which are located on the coasts of North, Southwest, and East China, were selected. A total of 35 districts or counties were randomly selected based on geographic location and population distribution. These districts or counties include: Xinhua District of Shijiazhuang City, Xinji County of Shijiazhuang City, Lubei District of Tangshan City, Ren County of Xingtai City, Neiqiu County of Xingtai City, Wuqiang County of Hengshui City, Wangdu County of Baoding City, Lianchi District of Baoding City, Daming County of Handan City, Zhangbei County of Zhangjiakou City, Shuangqiao District of Chengde City, Xinhua District of Cangzhou City, Nanming District of Guiyang City, Kaili City of Qiandongnan Miao and Dong Autonomous Prefecture of Guizhou City, Kaiyang County of Guiyang City, Danzhai County of Qiandongnan Miao and Dong Autonomous Prefecture, Shuicheng County of Liupanshui City, Dejiang County of Tongren City, Shibei District of Qingdao City, Penglai City of Yantai City, Laizhou City of Yantai City, Lijin County of Dongying City, Shouguang City of Weifang City, Wucheng County of Dezhou City, Lingcheng District of Dezhou City, Lanshan District of Linyi City, Donga County of Liaocheng City, Dingtao District of Heze City, Linzi District of Zibo City, Sishui County of Jining City, Yishui County, and Linyi City. Two townships (streets) were selected from each district and county, and two village (neighborhood) committees were selected from each township (street). In each village (neighborhood) committee, 15 lactating women aged 18–50 were selected, and all of them were within 2 years postpartum. Subjects with overt disease, such as chronic urinary system disease, end-stage renal disease, tumor, and organ transplant, were excluded. We gathered subjects in township health centers, village clinics, and community health service centers for investigation. For this investigation, China CDC organized experts to formulate unified work manuals on the operation methods and quality control methods of the questionnaire, physical examination, and other project links. In order to ensure the quality of work, each province set up provincial quality control groups, which are trained and assessed strictly by the national working group in advance. All links of the investigation, including training, questionnaire survey, physical examination, blood sample transfer, and laboratory testing, have been strictly controlled. During the survey, quality control surveys were conducted using tablet computers and computer programs, and subjects who were unable to participate were promptly replaced. Promptly replaced occurs in the sampling stage, and each survey point is required to not exceed 5% of the amount. The subjects were approached, contacted, screened, etc., through local CDC departments and community health service centers. Local CDC staff screened participants/subjects for eligibility to be included (or not eligible) for participation in this study. The inclusion and exclusion process is shown in Figure 1.

According to nutrition survey data over the years, the sample size of this survey was calculated with the anemia rate of Chinese lactating mothers in 2013 at 9.3%. If the relative standard error is within 15%, r = 15%, δ=15% × 9.3%, to ensure 1.395%. The confidence level was 95% (bilateral), i.e., μ = 1.96, to ensure accuracy. Accordingly, the sample size of the survey should be at least 1665.

The calculation formula:(1)$$n=\frac{\mu_{\alpha /2}^{2}*p(1-p)}{\delta^{2}}$$

This survey was approved by the Ethics Committee of the National Institute of Nutrition and Health and the Chinese Center for Disease Control and Prevention (number: 201614). Participants understood all aspects of the informed consent form and voluntarily participated in the survey. Meanwhile, all participants signed an informed consent form before participating in the project. 

This study included questionnaires, physical examinations, and laboratory tests. Working groups at the national and provincial level were responsible for the quality control of the surveys. District and county-level staff trained by the local CDC were responsible for fasting blood sample collection, interviews, and questionnaires. All participants underwent two steps: questionnaire survey and physical examination.

Inclusion criteria for all subjects were 18 ≤ age ≤ 50 years and BMI < 40 kg/m^2^. The number of individuals of 40 is very small in this survey, and if included, it will become an extreme value to interfere with the stability of the research results. This principle has also been applied in the previous literature [22]. The diagnostic criteria for dyslipidemia refer to the “Guidelines for the Prevention and Treatment of Dyslipids in Chinese Adults (Revised 2016)” [23], that is, hypercholesterolemia: total cholesterol (TC) ≥ 6.20 mmol/L; hypertriglyceridemia: triglyceride (TG) ≥ 2.30 mmol/L; high-LDL-cholesterolemia: LDL-C (low-density lipoprotein cholesterol) ≥ 4.10 mmol/L; low HDL-cholesterolemia: HDL-C (high-density lipoprotein cholesterol) < 1.00 mmol/L, any of which was called dyslipidemia. Since the population in this study was lactating women, there were no individuals in the sample who were previously known to have dyslipidemia. First of all, none of the pregnant women in our study were taking lipid-lowering drugs, and we determined dyslipidemia by measuring blood lipid levels in these subjects during the study. Regular physical examination for lactating women is not common in China. The diagnosis of dyslipidemia was completed when lipid laboratory measurements were completed. Basic characteristics, including age, sex, BMI, blood pressure, physical activity level, smoking status, drinking status, parity number, history of gestational hypertension, history of gestational diabetes, time of postpartum (age of breastfeeding child), etc., were obtained through standardized questionnaires designed by the national survey group. Anthropometric measurements, such as weight (kg) and height (cm), were performed by uniformly trained district/county CDC staff. The height and weight of the subjects were measured once by the investigator. Blood pressure was measured three times and averaged. All subjects underwent physical examinations in the early morning after an overnight fast, and they were asked to remove shoes and thick clothing and untie their hair buns during the measurement. BMI was calculated after the survey, and the formula was weight (kg)/height squared (m^2^). At the same time, the equipment used for anthropometric measurement is a unified model, electronic weight scale (G&G TC-200K) and electronic blood pressure monitor (OMRON HBP1300), with measurement accuracy of 0.1 cm, 0.1 kg and 1 mmHg, respectively. 

Light physical labor is defined as sitting 75% of the time and standing 25% of the time with special occupational activities office work, repairing electrical clocks, shop assistants, hotel attendants, chemical experiment operations, lectures, etc. Moderate physical labor is defined as sitting 25% of the time and standing 75% of the time in special occupational activities, student activities, motor vehicle driving, electrical installation, lathe operation, metalworking, etc.

### 2.2. Laboratory Measurement

Blood samples were collected from the median cubital vein of all subjects in township health centers, village clinics, and community health service centers. The blood samples were separated into plasma within 1 h, sent to the laboratory through the cold chain, and frozen at −80 °C for later use. Fasting plasma glucose (FPG), serum zinc, hemoglobin, serum vitamin B12, vitamin A, vitamin D, hs-CRP, transferrin receptor, ferritin, albumin, total protein and folic acid, total cholesterol (TC), triglycerides (TG), low-density lipoprotein cholesterol (LDL-C), and high-density lipoprotein cholesterol (HDL-C) were measured by a Hitachi Autoanalyzer 7600 (Hitachi, Tokyo, Japan). All measurements were performed by professional laboratory personnel, while a strict quality control was carried out in the laboratory. All questionnaires and physical measurements were trained by China CDC, and blood parameters were measured by China CDC laboratories.

### 2.3. Statistical Analysis

Logistic regression analysis (categorical variables) and generalized linear models (continuous variables) were used to assess differences in basic and clinical characteristics between different BMI subgroups. Basic characteristics included age, sex, body mass index (BMI), blood pressure, physical activity level, smoking status, drinking status, parity number, history of gestational hypertension, history of gestational diabetes, and time of postpartum (age of breastfeeding child). Generalized linear models were used to estimate the association between blood lipids and BMI. Multiple logistic regression analysis was used to estimate the relationship between BMI and risk of dyslipidemia. To estimate the odds ratios (ORs) for different types of dyslipidemia, BMIs were grouped by quartile: quartile 1 (Q1), <21.45 kg/m^2^; quartile 2 (Q2), 21.45 to <23.72 kg/m^2^; quartile 3 (Q3), 23.72 to <26.42; quartile 4 (Q4), ≥26.42. According to the quartile division, the sample distribution is more uniform, which can provide a better basis for verifying the trend. Data are presented as *n* (%) for categorical data, and mean (standard deviation, SD) for parametrically distributed data. Regression models were adjusted for potential confounding factors, including age, sex, blood pressure, physical activity level, smoking status, alcohol consumption status, parity numbers, history of gestational hypertension, history of gestational diabetes, and breastfeeding child months age. Since the overall sample size was greater than 2000, according to the central limit theorem, a properly normalized sum of independent random variables tends toward a normal distribution, even if the original variables are not normally distributed. In terms of sensitivity analysis, we conducted a stratified analysis by age, months of breastfeeding children, first birth or not, physical activity level, blood glucose, and hemoglobin. Meanwhile, interaction analyses were performed to assess the effect of stratification factors on the relationship between BMI and risk of dyslipidemia. We further explored the nonlinear relationship between BMI and ORs of dyslipidemia using restricted cubic splines (RCS) and selected 3 nodes for curve fitting according to the AIC optimality principle. BMI cut-off value, basing on the Youden index computed from ROC analysis, was used to explore BMI threshold with the smallest risk of dyslipidemia in lactating women [24]. Statistical analysis was performed using R 4.1.2. *p* < 0.05 (two-tailed) was considered significant. 

## 3. Results

### 3.1. Characteristics of Study Participants

The mean age of the subjects was 29.78 years. Among the 2295 lactating women subjects, the BMI was 24.21 ± 3.89 kg/m^2^. The average age of their lactating children was 7.86 ± 5.03 months. In these subjects, 792 (34.5%) lactating women gave birth once, and 1393 (60.7%) gave birth twice. There were 65 subjects with hypercholesterolemia, 122 with hypertriglyceridemia, 133 with hypo-HDL-cholesterolemia, and 73 subjects with hyper-LDL-cholesterolemia.

Table 1 lists the characteristics of study participants. Participants with higher BMI were more likely to be older, have more parity numbers, and have a history of gestational diabetes. With increasing BMI, levels of FPG, TC, TG, LDL-C, transferrin receptor, hemoglobin, hs-CRP, systolic pressure increased (*p* < 0.05). Conversely, serum vitamin A, HDL, vitamin B12, and folic acid levels decreased (*p* < 0.05).

Table 2 explores the linear relationship between BMI and blood lipids in the different children age groups. In the <12 month and 12–23 month age groups, BMI was positively correlated with TC, TG, and LDL-C (*p* < 0.05). However, in both groups, BMI was inversely correlated with HDL-C (*p* < 0.0001).

### 3.2. Logistic Regression Results

Table 3 shows the logistic regression results of the association between BMI and risk of all types of dyslipidemia in lactating women. After the model adjusted for confounding factors, such as age, parity numbers, breastfeeding child age (months), current smoking status, current drinking status, physical activity level, FPG, history of gestational diabetes, history of gestational hypertension, hemoglobin, ORs (95% CI) were statistically significant for dyslipidemia, hypercholesterolemia, hypertriglyceridemia, hypo-HDL-cholesterolemia and hyper-LDL-cholesterolemia. Higher BMI was associated with higher ORs for all types of dyslipidemia.

The OR (95% CI) of dyslipidemia was 3.828 (2.503, 5.853), hypercholesterolemia was 3.153 (1.253, 7.933), hypertriglyceridemia was 5.491 (2.537, 11.884), hypo-HDL-cholesterolemia was 3.692 (2.070, 6.584), and hyper-LDL-cholesterolemia was 3.091 (1.371, 6.967), by comparing the highest with lowest quartiles of BMI. The ORs of all types of dyslipidemia tended to increase with increasing BMI (*p* < 0.05).

Sensitivity analyses for dyslipidemia are concentrated in Table 4. The results showed that the positive association of BMI with risk of dyslipidemia in lactating women was nearly consistent across all stratified analyses. The associations were stable across age groups, breastfeeding child age (months), first birth (yes/no), physical activity level, FPG, and hemoglobin. These factors did not interact with this relationship (*p* > 0.05).

### 3.3. BMI and Risk of Dyslipidemia in Lactating Women

The sample distribution of various types of dyslipidemia in lactating women subjects is shown in Figure 2, and subjects with different types of dyslipidemia are presented. In RCS based on logistic regression models, the ORs of dyslipidemia increased significantly with increasing BMI in lactating women (Figure 3). The nonlinear spline test was not statistically significant (*p* nonlinear = 0.4673), indicating a linear relationship between BMI and risk of dyslipidemia in lactating women.

Figure 4 shows the ROC curve for the diagnosis of dyslipidemia using five indicators, AUC is BMI 0.655 (95% CI 0.621–0.688, *p* < 0.0001), TC 0.645 (95% CI 0.601–0.689, *p* < 0.0001), TG 0.864 (95% CI 0.838–0.891, *p* < 0.0001), HDL-C 0.188 (95% CI 0.155–0.220, *p* < 0.0001), and LDL-C 0.637 (95% CI 0.594–0.679, *p* < 0.0001). The best cut-off value for BMI was 24.85 kg/m^2^ according to the Youden index standard.

## 4. Discussion

To the best of our knowledge, this was the first large-scale population-based study of the association between BMI and risk of dyslipidemia in lactating women. We found that higher BMI in lactating women was associated with higher ORs of dyslipidemia (all types). Additional adjustments for potential confounders did not materially affect these results. BMI in lactating women was also positively correlated with TC, TG, LDL-C, but negatively correlated with HDL-C (*p* < 0.05). Using ROC analysis, we found the optimal cut-off point for BMI in lactating women was 24.85 kg/m^2^, which can distinguish the minimal risk of dyslipidemia.

Overweight and obesity in lactating women are public health issues of concern. Postpartum weight maintenance may predispose women to obesity [23]. It has been reported that up to 20% of women gain more than 5 kg of weight one year postpartum compared with pre-pregnancy [24,25]. Importantly, studies have shown that postpartum weight is associated with an increased risk of many adverse outcomes in subsequent pregnancies, independent of a woman’s initial BMI [26,27,28]. A study in the Mexican population showed that the children of postpartum overweight mothers had a higher tendency to be obese [29]. A study from an Australian population showed that women’s BMI decreased significantly between 2 and 12 months of breastfeeding, accompanied by other body composition changes such as decreased fat content [30]. This provides positive recommendation for postpartum women to maintain their ideal weight or lose weight. Because of the particularity of the physiological state of lactating women, it is of great practical significance to explore the relationship between BMI and dyslipidemia.

Our study was the first to verify the association of BMI with blood lipids and dyslipidemia in lactating women. Even after stratification by age of their breastfeeding children, BMI was still positively correlated with TC, TG, and LDL-C, but negatively correlated with HDL-C in these lactating women. Lactating women are a special group, and the transition during lactation represents huge physiological and immune changes in the human body. Previous studies have found that the changes in mRNA expression in the body during lactation affect a series of metabolic enzymes, resulting in corresponding changes in fat metabolism and lipid metabolism. Therefore, it is necessary to pay attention to the changes in BMI and blood lipid during lactation [31]. Therefore, research on special populations during lactation is of great significance to evidence-based medicine. Although there is currently a lack of historical research literature on the relationship between BMI and blood lipids in lactating women, previous literature has demonstrated that this association is common in other populations [32]. In a study on American adults, it was found that the main factor affecting blood lipid levels was BMI [33]. A study in a Sri Lankan adult population showed that serum TG level was positively correlated with BMI, while serum LDL-C was negatively correlated with BMI [34]. This was the exact opposite of what we have observed in LDL-C. A Norwegian population-based study analyzed blood lipids 3–6 months postpartum in women with hyper-cholesterolemia. Compared with 36 weeks of gestation in 14 subjects, the results showed that plasma TC and LDL-C were reduced by 23% and 25%, respectively, while plasma TG decreased by 72% at 3–6 months postpartum [35]. These results imply that the lipid metabolism in pregnant and lactating women has undergone great changes, and our study supplements the epidemiological data on biochemical associations in lactating women from one aspect.

Our findings suggested that BMI was positively associated with the risk of hypercholesteremia, hypertriglyceridemia, hypo-HDL-cholesterolemia, hyper-LDL-cholesterolemia, and dyslipidemia in lactating women. These relationships remained very stable after multiple confounder adjustments, stratified analyses, and nonlinear model exploration. The existing literature has observed similar phenomena in other populations. As the main component of metabolic syndrome, obesity was often accompanied by dyslipidemia [36,37,38,39,40,41]. At the same time, studies have shown that adult dyslipidemia was directly related to the increase in BMI [9,42]. In a cohort study in a Japanese population, women with BMI ≥ 25.0 kg/m^2^ had a 1.54-fold higher risk of dyslipidemia than women with BMI < 25.0 kg/m^2^ [43]. In addition, a previous i3C consortium analysis showed that increased childhood BMI affected adult blood lipids through adult obesity, with the strongest association between adult BMI and concurrent dyslipidemia [44]. In diabetic patients, with increasing BMI (*p* < 0.0001), the frequency of dyslipidemia gradually increased [10]. It can be seen that the relationship between BMI and dyslipidemia was widespread in various populations, and our study further verified this relationship in the lactating women population. Since the purpose of our study is to analyze the influence of BMI on dyslipidemia, due to the limited space of this paper, we believe that the OR determination of dyslipidemia on BMI could not be included. Perhaps this aspect will be studied by our team in the future in an appropriate study.

This study confirmed the stable linear relationship between BMI and the risk of dyslipidemia in lactating women, and suggested that lactating women should control BMI < 24.85 kg/m^2^ in order to prevent dyslipidemia. In this study, sensitivity analysis of dyslipidemia found that the positive association between BMI and dyslipidemia in lactating women was stable across all stratifications. There was no interaction with stratification factors, indicating that the relationship between BMI and dyslipidemia was not significantly affected by these factors. At the same time, the OR of dyslipidemia increased rapidly with the increasing BMI in a linear relationship. This also provided a stabilizing factor for exploring optimal BMI thresholds that minimize the risk of dyslipidemia in lactating women. In a 2005 Iranian national cross-sectional study, the cut-off point for female BMI was 26.1 kg/m^2^ based on the maximum sum of sensitivity and specificity for detecting hypercholesterolemia in females [45]. According to the recommendations of the World Health Organization, normal weight, overweight, and obesity are divided by <25, 25–30, and >30 kg/m^2^. In China, the BMI of overweight has the best sensitivity and specificity for the detection of chronic disease risk factors (hypertension, diabetes, dyslipidemia) with a reference range of 24 kg/m^2^, and the cut-off point for BMI to define obesity is 28 kg/m^2^ with a specificity of 90% [46]. In this study, aiming at detecting dyslipidemia, the Youden index criterion was used to select the best cut-off value for BMI of 24.85 kg/m^2^, through the area under curve (AUC) under receiver operating characteristic curve (ROC) and the sensitivity and specificity of different BMI cut-points. This is in line with the WHO criteria for overweight classification. The cut-off value of BMI in lactating women was proposed to verify whether it was significantly different from WHO. In our study, no significant difference was found, which was similar to the negative results. However, the previous literature reports on this aspect are rare. We maintain that it is necessary and meaningful in terms of the fundamental data on the human life cycle in this study. This cut-off value allows healthcare professionals to clarify the principles guiding nursing women to maintain a healthy weight.

Our study has several strengths. First, we used stricter inclusion criteria than general studies, that is, lactating women were all within 2 years postpartum, and still had breastfeeding behavior and a BMI < 40 kg/m^2^. It was difficult to find breastfeeding women on such a large scale in general studies. Second, strict quality control were conducted. We have developed a unified work manual and experimental manual, and conducted unified training for all investigators. At the same time, national and provincial quality control groups have been formed to track and control each link. Third, we tried to explore the optimal BMI threshold with the least risk of dyslipidemia in lactating women, which has practical guiding significance for disease prevention. Fourth, we explored the findings by hierarchical analysis, interaction analysis, and nonlinear analysis. These statistical methods were used to verify the dose–response relationship and stability of BMI, blood lipids, and dyslipidemia in lactating women.

We acknowledge that our research has some limitations. First, since our study was not a prospective cohort study, the findings were correlational, not causal. Second, this study lacks dietary data to adjust for the effects of diet. Second, physical activity is a key contributor to cardiometabolic health outcomes, but a validated questionnaire to assess physical activity levels was not used in this study. In addition, future research should explore the interplay between dietary intake, BMI, and risk of dyslipidemia in lactating women. 

In conclusion, our study observed that BMI was positively correlated with TC, TG, and LDL-C, but negatively correlated with HDL-C in lactating women. Meanwhile, higher BMI in lactating women was associated with higher risk of hypercholesteremia, hypertriglyceridemia, hypo-HDL-cholesterolemia, hyper-LDL-cholesterolemia, and dyslipidemia. The relationship between BMI and the OR of dyslipidemia was linear. BMI 24.85 kg/m^2^ was the best cut-off point for the prevention of dyslipidemia in lactating women.

## Figures and Tables

**Figure 1 nutrients-14-05174-f001:**
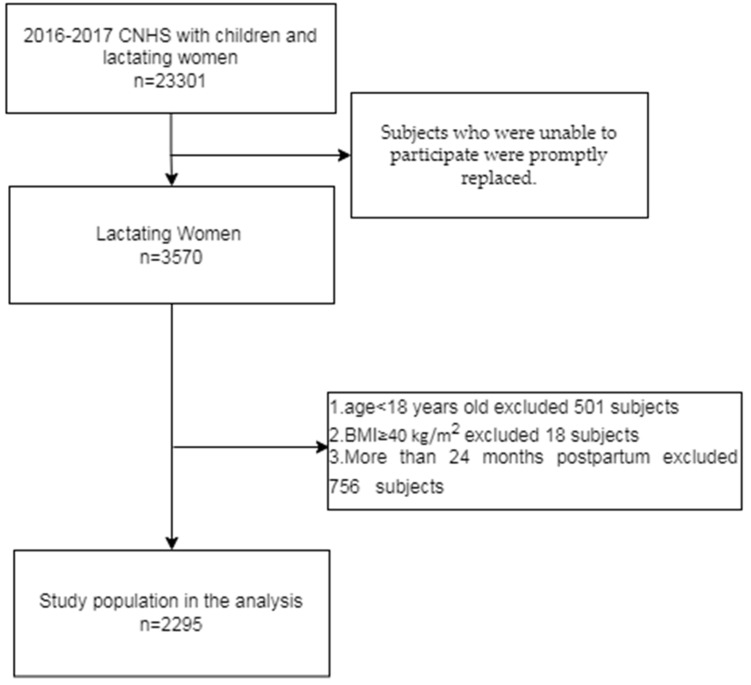
Inclusion and Exclusion Flowchart.

**Figure 2 nutrients-14-05174-f002:**
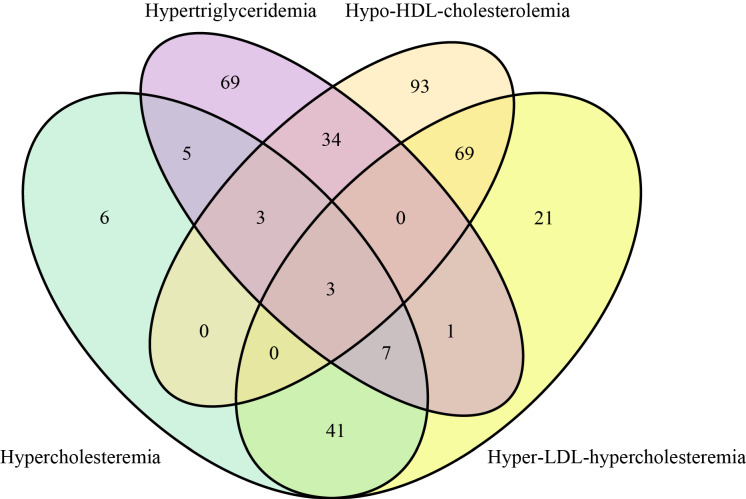
Venn diagram of sample distribution of four types of dyslipidemia subjects.

**Figure 3 nutrients-14-05174-f003:**
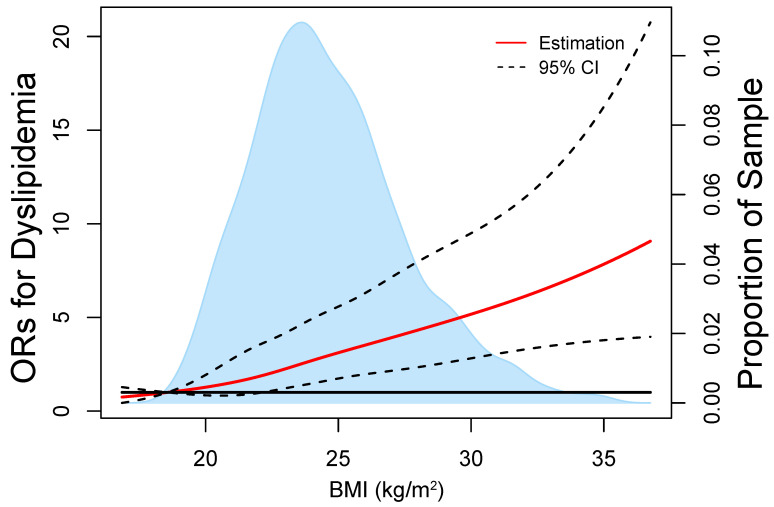
Representation of restricted cubic spline logistic regression models for BMI and risk of dyslipidemia. Red solid line shows OR as a function of BMI adjusted for age, age of children, numbers of birth, current smoking status, and current alcohol drinking.

**Figure 4 nutrients-14-05174-f004:**
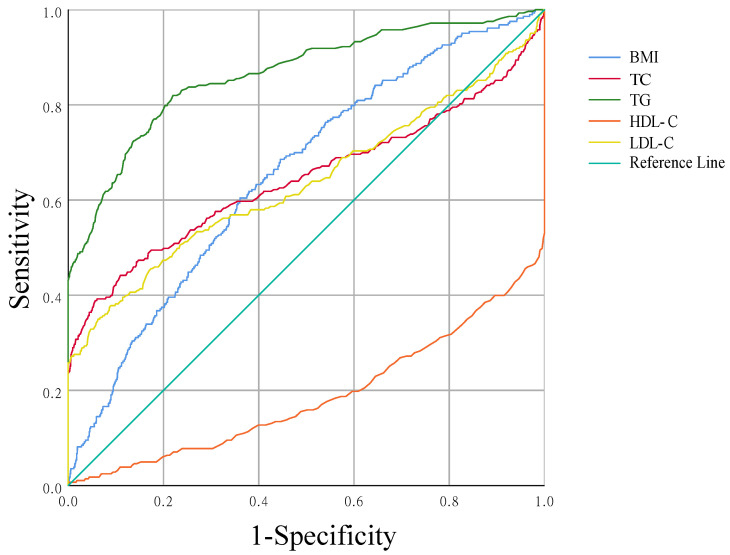
Receiver operator characteristic (ROC) curves for BMI, TC, TG, HDL-C, and LDL-C to predict dyslipidemia. Area under ROC curve = 0.655, 0.645, 0.864, 0.188, 0.637.

**Table 1 nutrients-14-05174-t001:** Characteristics by BMI quartile in lactating women.

	Quartile of BMI (kg/m^2^)				F/χ^2^	*p*
	Q1 (Reference), <21.45 (*n* = 574)	Q2, 21.45 to <23.72 (*n* = 574)	Q3, 23.72 to <26.42 (*n* = 573)	Q4, >26.42 (*n* = 574)		
Age (years)	28.2 (4.77)	29.7 (5.09)	30.3 (5.44)	30.9 (5.58)	9.93	0.0016
BMI	19.72 (1.26)	22.60 (0.65)	25.04 (0.78)	29.48 (2.58)	8222.1	<0.0001
Parity numbers	1.60 (0.58)	1.71 (0.58)	1.76 (0.56)	1.78 (0.56)	7.71	0.0055
Age of children (months)	8.62 (5.17)	7.51 (4.74)	7.69 (5.02)	7.62 (5.10)	11.22	0.0008
Drinker (%)	10 (1.7)	11 (1.9)	13 (2.3)	9 (1.6)	0.4559	0.4996
Smoker (%)	8 (1.4)	1 (0.2)	6 (1.1)	3 (0.5)	0.1352	0.7131
Physical activity level						
Light	474 (83.60)	468 (81.68)	454 (80.78)	449 (80.04)	0.6511	0.4197
Moderate	89 (15.70)	99 (17.28)	102 (18.15)	112 (19.96)		
Vigorous	4 (0.71)	6 (1.05)	6 (1.07)	0		
History of gestational diabetes (%)	16 (2.8)	20 (3.5)	28 (4.9)	39 (6.8)	5.665	0.0173
History of gestational hypertension (%)	8 (1.4)	15 (2.6)	15 (2.6)	37 (6.5)	1.482	0.2235
TC (mmol/L)	4.13 (0.74)	4.26 (0.84)	4.42 (0.9)	4.50 (0.93)	22.18	<0.0001
TG (mmol/L)	0.81 (0.5)	0.98 (0.64)	1.13 (0.71)	1.35 (0.87)	72.11	<0.0001
HDL (mmol/L)	1.67 (0.37)	1.53 (0.34)	1.46 (0.34)	1.36 (0.31)	145.52	<0.0001
LDL (mmol/L)	2.25 (0.65)	2.44 (0.76)	2.60 (0.8)	2.70 (0.84)	57.05	<0.0001
Serum zinc (μg/dL)	83.18 (17.94)	83.39 (23.21)	83.75 (23.54)	84.63 (23.35)	1.16	0.2813
Blood glucose (mmol/L)	4.68 (0.59)	4.80 (0.58)	4.88 (0.73)	5.11 (1.2)	28.7	<0.0001
Ferritin (ng/mL)	47.91 (36.25)	47.99 (41.77)	52.84 (47.69)	53.48 (41.53)	1.54	0.2155
Transferrin receptor (mg/L)	3.20 (1.32)	3.36 (2.11)	3.44 (1.86)	3.50 (1.60)	15.26	<0.0001
Hemoglobin (g/L)	130.69 (12.20)	130.96 (12.59)	131.77 (13.94)	132.98 (14.28)	17.61	<0.0001
Serum Vitamin B12 (pg/mL)	469.70 (236.76)	454.13 (245.92)	449.64 (238.22)	444.42 (209.6)	9.76	0.0018
Serum folic acid (ng/mL)	6.38 (3.54)	6.23 (3.52)	6.16 (3.57)	5.53 (3.08)	12.10	0.0005
Hs-CRP (mg/L)	1.17 (3.96)	1.51 (3.83)	1.72 (3.94)	2.70 (4.32)	25.05	<0.0001
Albumin (g/L)	48.06 (3.16)	47.50 (3.29)	47.56 (3.73)	47.49 (3.22)	13.88	0.0002
Total protein (g/L)	75.82 (4.89)	75.69 (5.27)	76.12 (5.31)	76.24 (4.98)	2.00	0.1571
Serum Vitamin A (µmol/L))	0.46 (0.34)	0.47 (0.14)	0.50 (0.28)	0.50 (0.16)	7.51	0.0062
Serum Vitamin D (ng/mL)	17.78 (7.13)	17.23 (5.83)	17.32 (5.9)	17.38 (5.98)	0.25	0.6164
Systolic pressure (mmHg)	112.37 (11.20)	114.49 (10.54)	117.61 (10.68)	122.81 (12.67)	87.29	<0.0001
Diastolic pressure (mmHg)	68.65 (8.95)	70.76 (8.94)	72.52 (8.79)	75.67 (10.19)	2.70	0.1005

Data are presented as *n* (%) for categorical data, mean (standard deviation, SD) for parametrically distributed data. Chi-square test (categorical variables) and generalized linear models (continuous variables) were used to assess differences in basic and clinical characteristics between different BMI subgroups.

**Table 2 nutrients-14-05174-t002:** BMI values according to age of children (months) and BMI quartile in lactating women (mean values and standard deviations).

Age of Children (Months)	Q1 (Reference), <21.45 (*n* = 574)	Q2, 21.45 to <23.72 (*n* = 574)	Q3, 23.72 to <26.42 (*n* = 573)	Q4, >26.42 (*n* = 574)	F Value	*p*
<12 months	*n* = 431	*n* = 473	*n* = 467	*n* = 460		
TC (mmol/L)	4.19 (0.78)	4.31 (0.82)	4.46 (0.92)	4.56 (0.95)	15.07	0.0001
TG (mmol/L)	0.80 (0.41)	0.99 (0.64)	1.14 (0.72)	1.36 (0.87)	57.48	<0.0001
HDL-C (mmol/L)	1.68 (0.38)	1.53 (0.34)	1.48 (0.34)	1.37 (0.31)	102.28	<0.0001
LDL-C (mmol/L)	2.29 (0.67)	2.47 (0.75)	2.63 (0.83)	2.74 (0.86)	38.93	<0.0001
12–23 months	*n* = 143	*n* = 101	*n* = 106	*n* = 114		
TC (mmol/L)	3.96 (0.59)	4.07 (0.93)	4.25 (0.78)	4.27 (0.82)	4.37	0.0372
TG (mmol/L)	0.83 (0.71)	0.94 (0.62)	1.09 (0.63)	1.31 (0.89)	14.76	0.0001
HDL-C (mmol/L)	1.62 (0.34)	1.5 (0.3)	1.41 (0.32)	1.31 (0.29)	40.82	<0.0001
LDL-C (mmol/L)	2.13 (0.56)	2.28 (0.8)	2.5 (0.69)	2.54 (0.7)	12.39	0.0005

Statistics were performed using generalized linear models. Models were adjusted for age, serum zinc, age of children, parity number, history of gestational diabetes, history of gestational hypertension, blood pressure, fasting blood glucose, hemoglobin, serum vitamin B12, vitamin A, vitamin D, hs-CRP, transferrin receptor, ferritin, albumin, total protein, and folic acid.

**Table 3 nutrients-14-05174-t003:** ORs (95% CI) of different types of dyslipidemia by quartiles of BMI in lactating women.

	Q1 (Reference), <21.45 (*n* = 574)	Q2, 21.45 to < 23.72 (*n* = 574)	Q3, 23.72 to < 26.42 (*n* = 573)	Q4, >26.42 (*n* = 574)	χ^2^	*p* for Trend
Hypercholesterolemia						
Case/control subjects, *n*	6/568	10/564	20/553	29/545		
Crude OR (95% CI)	1	1.68 (0.61, 4.66)	3.42 (1.37, 8.59)	5.04 (2.08, 12.23)	17.85	0.0005
Adjusted OR * (95% CI)	1	1.57 (0.56, 4.35)	2.85 (1.12, 7.29)	4.48 (1.82, 11.01)	15.10	0.0017
Adjusted OR † (95% CI)	1	1.38 (0.50, 3.87)	2.35 (0.91, 6.09)	3.15 (1.25, 7.93)	8.45	0.0376
Hypertriglyceridemia						
Case/control subjects, *n*	8/566	27/547	35/538	52/522		
Crude OR (95% CI)	1	3.50 (1.58, 7.77)	4.60 (2.12, 10.01)	7.05 (3.32, 14.98)	28.90	<0.0001
Adjusted OR * (95% CI)	1	3.32 (1.49, 7.40)	4.28 (1.96, 9.38)	6.84 (3.2, 14.63)	27.93	<0.0001
Adjusted OR † (95% CI)	1	3.17 (1.42, 7.06)	3.93 (1.79, 8.64)	5.49 (2.54, 11.88)	19.92	0.0002
Hypo-HDL-cholesterolemia						
Case/control subjects, *n*	17/557	24/550	39/534	53/521		
Crude OR (95% CI)	1	1.43 (0.76, 2.70)	2.39 (1.34, 4.28)	3.33 (1.91, 5.83)	23.05	<0.0001
Adjusted OR * (95% CI)	1	1.49 (0.79, 2.82)	2.56 (1.42, 4.6)	3.66 (2.08, 6.45)	25.78	<0.0001
Adjusted OR † (95% CI)	1	1.59 (0.84, 3.01)	2.66 (1.48, 4.8)	3.69 (2.07, 6.58)	24.10	<0.0001
Hyper-LDL-cholesterolemia						
Case/control subjects, *n*	8/566	11/563	21/552	33/541		
Crude OR (95% CI)	1	1.38 (0.55, 3.47)	2.69 (1.18, 6.12)	4.31 (1.98, 9.42)	19.43	0.0002
Adjusted OR * (95% CI)	1	1.29 (0.52, 3.25)	2.40 (1.04, 5.52)	3.88 (1.75, 8.58)	16.70	0.0008
Adjusted OR † (95% CI)	1	1.20 (0.48, 3.03)	2.15 (0.92, 4.99)	3.09 (1.37, 6.97)	11.18	0.0108
Dyslipidemia						
Case/control subjects, *n*	32/542	54/520	82/491	115/459		
Crude OR (95% CI)	1	1.76 (1.12, 2.77)	2.83 (1.85, 4.33)	4.24 (2.81, 6.40)	57.51	<0.0001
Adjusted OR * (95% CI)	1	1.72 (1.09, 2.71)	2.68 (1.74, 4.12)	4.20 (2.77, 6.37)	55.31	<0.0001
Adjusted OR † (95% CI)	1	1.70 (1.08, 2.69)	2.61 (1.69, 4.02)	3.83 (2.50, 5.85)	45.73	<0.0001

* Model 1: adjusted for age, physical activity level, current smoking status, and current alcohol drinking status. † Model 2: adjusted for Model 1, age of children, numbers of birth, history of gestational diabetes, history of gestational hypertension, fasting blood glucose, and hemoglobin.

**Table 4 nutrients-14-05174-t004:** Stratified analyses of dyslipidemia risk and BMI by age, age of children, firstborn (yes/no), physical activity level, blood glucose, and hemoglobin.

	Q1 (Reference), <21.45 (*n* = 574)	Q2, 21.45 to <23.72 (*n* = 574)	Q3, 23.72 to <26.42 (*n* = 573)	Q4, >26.42 (*n* = 574)	χ^2^	*p* for Trend	*p* Value for Interaction
Age (years)							0.9138
<30	1	1.93 (1.04, 3.57)	3.16 (1.75, 5.70)	4.43 (2.48, 7.94)	28.6503	<0.0001	
≥30	1	1.31 (0.66, 2.6)	1.99 (1.04, 3.80)	3.01 (1.61, 5.62)	17.4998	0.0006	
Age of children (months)							0.5652
<12 months	1	1.62 (0.96, 2.73)	2.25 (1.36, 3.72)	3.69 (2.26, 6.00)	33.9332	<0.0001	
12–23 months	1	2.08 (0.75, 5.76)	4.09 (1.66, 10.08)	4.05 (1.59, 10.34)	11.883	0.0078	
Parity							0.7659
Primiparous	1	1.83 (0.92, 3.65)	2.87 (1.47, 5.61)	4.57 (2.38, 8.79)	23.1572	<0.0001	
Multiparous	1	1.58 (0.85, 2.93)	2.41 (1.35, 4.31)	3.43 (1.94, 6.05)	23.1981	<0.0001	
Physical activity level							0.2562
Light	1	1.44 (0.86, 2.4)	2.61 (1.63, 4.2)	3.59 (2.25, 5.73)	38.0361	<0.0001	
Moderate & Vigorous	1	3.58 (1.16, 11.07)	3.14 (1, 9.88)	6.80 (2.25, 20.54)	12.4692	0.0059	
Fasting Blood glucose (mmol/L)							0.3006
<4.81 (median)	1	2.88 (1.51, 5.47)	3.05 (1.61, 5.81)	4.72 (2.47, 9.03)	22.0424	<0.0001	
≥4.81 (median)	1	0.97 (0.50, 1.90)	2.20 (1.22, 4.00)	3.12 (1.77, 5.52)	28.1374	<0.0001	
Hemoglobin (g/L)							0.7389
<132 (median)	1	2.28 (1.14, 4.56)	3.08 (1.56, 6.05)	5.41 (2.79,10.48)	27.5986	<0.0001	
≥132 (median)	1	1.26 (0.68, 2.34)	2.21 (1.25, 3.91)	2.80 (1.60,4.89)	18.7677	0.0003	

Data were OR (95% CI). The multivariate model was adjusted for age, age of children, numbers of birth, current smoking status, current alcohol drinking status, physical activity level, history of gestational diabetes, history of gestational hypertension, fasting blood glucose, and hemoglobin.

## Data Availability

The data are not allowed to be disclosed according to the National Institute for Nutrition and Health, Chinese Center for Disease Control and Prevention.

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
