# Peer review of "The Effect of BMI on Blood Lipids and Dyslipidemia in Lactating Women"

_nutrients, 2022, doi:10.3390/nu14235174_

Round 1

Reviewer 1 Report

In this manuscript, the authors present a study on the effect of BMI on blood lipids and dyslipidemia in lactating women in China.

Overall, the methodology used in this study is sound and is adequately described. The study design is in line with the research question and the results are clearly presented.

However, there are several comments the authors should address.

1) Language
The text would benefit from a thorough language revision. There are numerous grammar and spelling mistakes still present in the text.
The authors should use present simple when stating generally accepted facts or general truths (e.g. lines 48, 54, 55, 72, etc.)
The authors should check the use of the word "overweight" - it seems to me that in the text it is used as a noun (smilar to "obesity"), however, in English this is an adjective. Prehaps the authors meant "excess weight"?
The authors should check the senteces where they utilize the word "while" - this should be used to connect and contrast two opposing statements, not at the beginning of a sentence (lines 48 and 156). 
The sentences in lines 42 - 43, 58 - 59, and 191 are unclear and should be reworded.
The sentence in lines 148 - 149 should be reworded: "Among the subjects, 1845 (81.53%) experienced light physical labor, while 403 (17.76%) experienced moderate physical labor.
Figure 1 - the notation in the lower left corner states "hypercholesteremia" instead of "hypercholesterolemia"
There are also a lot of technical errors - lack of spaces between words and brackets (e.g. Table 1, Table 2, Table 3, Table 4 lines 165, 184, 192, 212 as well as extra spaces (e.g. lines 123, 154, 155, 173).

2) Introduction
The authors should expand the background in the introduction on why lactation could affect the relation between BMI and dyslipidemia. A longer elaboration on lines 54 - 59 is needed to lay a proper foundation on why the study was conducted. 

3) Materials and Methods
Line 111 - the term "antecubital vein" is not a proper anatomical term. The authors should utilize the proper anatomical terminology in scientific communication. I presume the authors were referring to the median cubital vein (v. mediana cubiti)?
Lines 131 - 133: The interpretation of the central limit theorem stated here is not entirely accurate. The central limit theorem states that the properly normalized sum of independent random variables tends toward a normal distribution, even if the original variables are not normally distributed. Or, in other words, the distribution of sample means approximates a normal distribution as the sample size gets larger, regardless of the population's distribution. This is a bit different from stating that the distribution of each variable is close to a normal distribution due to a large sample size (this is not necessarily true).
A brief explanation of the the criteria for light physical labor and moderate physical labor is necessary, since these terms are later used in the Results section.

4) Results
Figure 2 - the label for the right y-axis is a bit confusing. What does "fraction of population" mean in this context? Is this the proportion of subject involved in the study or does it refer to the entire population of lactating women in China? This should be clarified a better label should be found (e.g. consider using the term proportion rather than fraction).

5) Discussion
Overall, I feel that a clear explanation on the importance of this study is still missing and the conclusions seem a bit underwhelming considering the scope of the study.
The fact that increased BMI presents a risk for dyslipidemia is a well-known fact, as is the fact that lactation is a specific period accompanied by important physiological changes. It would be crucial to add a paragraph providing a comparison between the risk/association between BMI and dyslipidemia found in lactating women and the risk/association found in the general population/men/non-lactating women. This would show why the data in this study are notable. 
In addition, I am left wondering why the cut-off points for BMI presented in this study are significant, since they do not seem to differ much from the WHO criteria. The authors should comment on this in the Discussion section.

Author Response

1) Language

lines 48, 54, 55, 72, etc.

The text would benefit from a thorough language revision. There are numerous grammar and spelling mistakes still present in the text.

Thank you for your suggestion. We have found native English speakers for careful revision.

The authors should use present simple when stating generally accepted facts or general truths (e.g. lines 48, 54, 55, 72, etc.)

Thank you very much for your thoughtful and constructive suggestions. We have revised the grammar and spelling as you suggested in the revised manuscript.

The authors should check the use of the word "overweight" - it seems to me that in the text it is used as a noun (smilar to "obesity"), however, in English this is an adjective. Prehaps the authors meant "excess weight"?

Thank you for your review comments. In this paper, overweight is a clear definition concept, which has been widely used in many literatures. According to the recommendations of World Health Organization, normal weight, overweight and obesity were divided by <25, 25-30 and >30 kg/m2. In China, the BMI of overweight has the best sensitivity and specificity for the detection of chronic disease risk factors (hypertension, diabetes, dyslipidemia) with a reference range of 24 kg/m2 , and the cut-off point for BMI to define obesity was 28 kg/m2 with a specificity of 90%.

16.Winkvist, A.; Bertz, F.; Ellegård, L.; Bosaeus, I.; Brekke, H.K. Metabolic risk profile among overweight and obese lactating women in Sweden. PLoS One. 2013, 8, e63629.

22.Dalrymple, K.V.; Flynn, A.C.; Relph, S.A.; O'Keeffe, M.; Poston, L. Lifestyle Interventions in Overweight and Obese Pregnant or Postpartum Women for  Postpartum Weight Management: A Systematic Review of the Literature. Nutrients. 2018, 10.

33.Somasundaram, N.; Ranathunga, I.; Gunawardana, K.; Ahamed, M.; Ediriweera, D.; Antonypillai, C.N.; Kalupahana, N. High Prevalence of Overweight/Obesity in Urban Sri Lanka: Findings from the  Colombo Urban Study. J. Diabetes Res. 2019, 2019, 2046428.

45.Zhou, B.F. Predictive values of body mass index and waist circumference for risk factors of  certain related diseases in Chinese adults--study on optimal cut-off points of  body mass index and waist circumference in Chinese adults. Biomed. Environ. Sci. 2002, 15, 83-96.

The authors should check the senteces where they utilize the word "while" - this should be used to connect and contrast two opposing statements, not at the beginning of a sentence (lines 48 and 156). 

Thank you very much for your thoughtful and constructive suggestions. We have revised these according to your suggestion in the revised manuscript.

The sentences in lines 42 - 43, 58 - 59, and 191 are unclear and should be reworded.

Thank you for your review comments. We have revised and rewritten the words in the original text

The sentence in lines 148 - 149 should be reworded: "Among the subjects, 1845 (81.53%) experienced light physical labor, while 403 (17.76%) experienced moderate physical labor.

Thank you very much for your thoughtful and constructive suggestions. We have revised the sentence as you suggested in the revised manuscript.

Figure 1 - the notation in the lower left corner states "hypercholesteremia" instead of "hypercholesterolemia"

Thank you very much for your thoughtful and constructive suggestions. We have revised the word as you suggested in the revised manuscript.

There are also a lot of technical errors - lack of spaces between words and brackets (e.g. Table 1, Table 2, Table 3, Table 4 lines 165, 184, 192, 212 as well as extra spaces (e.g. lines 123, 154, 155, 173).

Thank you for your meticulous review. We have carefully modified it according to your requirements.

2) Introduction

The authors should expand the background in the introduction on why lactation could affect the relation between BMI and dyslipidemia. A longer elaboration on lines 54 - 59 is needed to lay a proper foundation on why the study was conducted. 

Thank you for your constructive suggestions. We have added explanations to the original text.

3) Materials and Methods

Line 111 - the term "antecubital vein" is not a proper anatomical term. The authors should utilize the proper anatomical terminology in scientific communication. I presume the authors were referring to the median cubital vein (v. mediana cubiti)?

Thank you very much for your thoughtful and constructive suggestions. We have revised the term as you suggested in the revised manuscript.

Lines 131 - 133: The interpretation of the central limit theorem stated here is not entirely accurate. The central limit theorem states that the properly normalized sum of independent random variables tends toward a normal distribution, even if the original variables are not normally distributed. Or, in other words, the distribution of sample means approximates a normal distribution as the sample size gets larger, regardless of the population's distribution. This is a bit different from stating that the distribution of each variable is close to a normal distribution due to a large sample size (this is not necessarily true).

Thank you for your professional guidance. We have used more professional expressions according to your guidance.

A brief explanation of the the criteria for light physical labor and moderate physical labor is necessary, since these terms are later used in the Results section.

Thank you for your rigorous guidance. We have added relevant content in the original text according to your guidance.

4) Results

Figure 2 - the label for the right y-axis is a bit confusing. What does "fraction of population" mean in this context? Is this the proportion of subject involved in the study or does it refer to the entire population of lactating women in China? This should be clarified a better label should be found (e.g. consider using the term proportion rather than fraction).

Thank you for your detailed review and constructive comments. We have modified the original image. It is the proportion of subject involved in the study.

5) Discussion

Overall, I feel that a clear explanation on the importance of this study is still missing and the conclusions seem a bit underwhelming considering the scope of the study.

The fact that increased BMI presents a risk for dyslipidemia is a well-known fact, as is the fact that lactation is a specific period accompanied by important physiological changes. It would be crucial to add a paragraph providing a comparison between the risk/association between BMI and dyslipidemia found in lactating women and the risk/association found in the general population/men/non-lactating women. This would show why the data in this study are notable. 

In addition, I am left wondering why the cut-off points for BMI presented in this study are significant, since they do not seem to differ much from the WHO criteria. The authors should comment on this in the Discussion section.

Thank you for your instructive advice. We have revised the article according to your comments. At the same time, references 9, 10, 31-33, 41, 43 in the original text also address your concerns. Meanwhile, The cut-off value of BMI in lactating women was proposed to verify whether it was significantly different from WHO. In our study, no significant difference was found, which was similar to the negative results. However, the previous literature reports on this aspect were rare. We still maintain that there is nothing new to be found in this study, but it is necessary and meaningful in terms of the fundamental data on the human life cycle.

Reviewer 2 Report

This study presents a positive association between BMI and fasting blood lipids (total cholesterol, triglycerides, LDL-cholesterol) and a negative association between BMI and HDL-cholesterol in lactating women. Higher BMI in lactating women was associated with a higher risk of various dyslipidaemias and in this cohort of lactating women, a BMI of 24.85kg/m2 appears to be the cut-off point for the prevention of dyslipidaemia. The authors present a number of results with sound statistical methodology; however, the scientific rational and critical discussion is lacking in rigor; significant detail in study methodology is also lacking and the fact that dietary intake was not measured nor reported remains a significant limitation to this research.

Overall – English grammar needs significant improvement in the entire manuscript. Many sentences don’t flow, unusual choice of words used in many places, some sentences abruptly end, incorrect tense often used and some basic punctuations are missing. Formatting of references need special attention as there are many inconsistencies in how authors, journals and types of evidence are referenced, intext citation brackets are also inconsistent.

Abstract

1.       Add space before (BMI) in line 19

2.       Define FPG in first instance

Introduction

1.       Overall, very short and limited in detail in terms of providing a strong rationale for this project.

2.       The English tense used in the introduction doesn’t fit. Past tense such as ‘was’ is used everywhere when referring to findings about linking BMI and dyslipidaemia. For e.g., the first line shouldn’t be past tense i.e., change to ‘body mass index IS a common measure…’ same in lines 40 & 41, 48 and others. Sentences appear unfinished e.g., line 48

3.       Literature presented in the first paragraph of introduction requires more elaboration. For e.g., line 46 should outline what that previous literature is in relation too and how is this relevant to your project?

4.       Be clear what you mean by ‘frequency of dyslipidaemia gradually increased with increasing BMI’ in line 46-7 and provide a little more information on the study details e.g., is this a meta-analysis? RCT? Humans? Men and women? Conditioned? Etc This level of detail should be in the entire introduction.

5.       Referring to lactation as a ‘special period’ is unscientific. Please be clear what you are actually trying to say here (lines 54-59).

6.       Provide information on what is known with respect to lactating women and cardiometabolic diseases, even if it’s in other populations; to help provide insight and rationale to your project.

7.       It hasn’t been convincingly justified why this phase of life (lactating) is critical to determine the optimal BMI range/upper limit with respect to dyslipidaemia. Suggest adding more information around the potential physiological implications (or even hypotheses) of excessive fat tissue (and discuss this in relation to body fat tissue / central obesity) to both the mother and baby; present what is the physiological link between BMI and dyslipidaemia; and what about other related cardiometabolic markers?

Methods:

Overall, a lot more detail is required in relation to the study population and data collection.

1.       Provide information as to why this survey was used and describe the cohort/population chosen to which the survey was implemented on

2.       Please state what Hebei, Guizhou and Shandong are i.e., are these cities? Regions?

3.       Is this sample representative of Chinese women of the same age? If so, provide evidence that this is a representative sample

4.       Out of the 2295 included, how many were not included? What total pool/sample/cohort have these women been drawn from? Please include a participant flow diagram to outline how you reached this number.

5.       What does ‘promptly replaced’ mean and by whom and how were participants replaced? This should be incorporated into a flow diagram.

6.       Please provide the following details regarding study population:

a.       How were participants recruited and by whom?

b.       Who randomly selected the districts?

c.       Who screened them for eligibility?

d.       Please report clearly how participants were to know they had dyslipidaemia? Were they screened prior to inclusion or were they supposed to know this beforehand somehow?

7.       Please provide the following details regarding data collection:

a.       It is unclear where all the data collection took place. Was this at a research facility? Out-patient clinic settings? Please provide this information.

b.       Why were other biochemical parameters such as vitamins and CRP measured? This should be justified or explained somewhere.

c.       Was weight, height, blood pressure collected serially? If so, please report

d.       Provide details on the specific questionnaires used (e.g., for physical activity, diet) and if they were validated where relevant

e.       Report how many times did participants visit the facility for data collection

f.        Where the blood was collected

Statistics

1. Please be specific re what ‘basic’ characteristics is referring too in lines 122

 2. Please outline in the methods section the significance and relevance of the BMI quartiles you have chosen to use in lines 126-127

3. Please refer to presenting means, SD or SEM etc – as you present results in this manner but haven’t stated this in the statistical methods section

Results

1.       Line 145, remove capital L on lactating

2.       Correct %’s and age to just 1 decimal place

3.       Line 152 should say ‘BMI quartiles’, line 154 correct to ‘with increasing BMI, levels of …’

4.       Please ensure all abbreviations have been defined in first instance

5.       Table 1, 2 3, 4 – typo for Q1 should be ‘reference’

6.       Include adequate description of statistical tests used in Table 1 footnote

7.       Table 1 title – suggest different wording. How is this ‘baseline’ when there is only one timepoint for this study?

8.       Table 3 & 4, present data to 2 decimal places

9.       Table 3 – It would be clinically relevant to also understand the OR for women who had multiple types of dyslipidaemias to see if their OR was higher in those with higher BMI. Can you provide this information?

Discussion

Please check the whole manuscript, and particularly Introduction and Discussion for correct English and grammar. Some punctuations like full stops are also missing.

1.       Please reword line 226, not correct grammar

2.       Need full stop in line 227.

3.       Please be consistent with in-text referencing. In the introduction you use [ ] but in the Discussion you use ( ).

4.       Please use scientific language for e.g., ‘positive tips’ in line 235; what are you trying to say here? Should be referring to the clinical translation of these findings in practice.

5.       Please include some discussion around the potential mechanisms of action to which you think might explain the relationship you’ve observed between BMI and dyslipidaemia. Is there anything inherent about lactation itself that may predispose women to have a higher BMI, fasting lipids or both? These areas should be discussed in light of your findings. Sections 236-253 and 254-267 only report associations/correlations, but these should be critically discussed and related back to the physiological mechanisms that might be at play here as well as the relevance to the lactation phase of a woman’s life.

6.       Correct spelling for ‘implied’ in line 251

7.       Line 269 – please refer to the BMI classification range, not just the cut points in lines 281 to 282. i.e., are you saying that in China, the BMI for overweight is over 24kg/m2?

8.       It should be explained in the methods or discussion why BMI of 40 or higher were excluded

9.       Lines 290-292 are vague. Please describe in detail what ‘unified work manuals’ are and what types of ‘unified training’ was required, what is ‘provincial quality control groups’ and what are the ‘links’ you’re referring too? These can be added to the methods section.

10.   The lack of dietary data is a significant limitation of this study and should be considered. Diet plays a heavy role in cardiometabolic risk factors such as BMI and dyslipidaemia, and moreover, dietary recommendations are different during the lactation life stage; which could influence both BMI and dyslipidaemia independently and interchangeably. It should be explained why dietary data was not collected and the potential impact of dietary intake should be discussed in the discussion.

11.   Include how these findings translate into practice, and how now knowing this cut off point of 24.85kg/m2 could inform care and management of lactating women by health care professionals.

Author Response

Abstract

  1. Add space before (BMI) in line 19

Thank you very much for your thoughtful and constructive suggestions. We have revised the space as you suggested in the revised manuscript.

  1. Define FPG in first instance

Thank you very much for your thoughtful and constructive suggestions. We have defined FPG in first instance (line 30) as you suggested in the revised manuscript.

Introduction

  1. Overall, very short and limited in detail in terms of providing a strong rationale for this project.

Thank you for your guiding suggestion. We have modified and added the corresponding content in the introduction according to your opinion.

  1. The English tense used in the introduction doesn’t fit. Past tense such as ‘was’ is used everywhere when referring to findings about linking BMI and dyslipidaemia. For e.g., the first line shouldn’t be past tense i.e., change to ‘body mass index IS a common measure…’ same in lines 40 & 41, 48 and others. Sentences appear unfinished e.g., line 48

Thank you very much for your thoughtful and constructive suggestions. We have revised the grammar as you suggested in the revised manuscript.

  1. Literature presented in the first paragraph of introduction requires more elaboration. For e.g., line 46 should outline what that previous literature is in relation too and how is this relevant to your project?

 Thanks for your enlightening guidance, we have arranged and modified the article paragraphs according to your suggestions.

  1. Be clear what you mean by ‘frequency of dyslipidaemia gradually increased with increasing BMI’ in line 46-7 and provide a little more information on the study details e.g., is this a meta-analysis? RCT? Humans? Men and women? Conditioned? Etc This level of detail should be in the entire introduction.

Thank you for your review. We have added more details of the literature according to your suggestion.

  1. Referring to lactation as a ‘special period’ is unscientific. Please be clear what you are actually trying to say here (lines 54-59).

Thank you for your review. We have carefully modified according to your comments and added part of the content in this paragraph to make a supplementary explanation.

  1. Provide information on what is known with respect to lactating women and cardiometabolic diseases, even if it’s in other populations; to help provide insight and rationale to your project.

 Thank you for your review. We have carefully modified according to your comments and added part of the content in this paragraph to make a supplementary explanation.

  1. It hasn’t been convincingly justified why this phase of life (lactating) is critical to determine the optimal BMI range/upper limit with respect to dyslipidaemia. Suggest adding more information around the potential physiological implications (or even hypotheses) of excessive fat tissue (and discuss this in relation to body fat tissue / central obesity) to both the mother and baby; present what is the physiological link between BMI and dyslipidaemia; and what about other related cardiometabolic markers?

 Thanks for your guidance, we have added corresponding literature and descriptions in the original text according to your comments.

Methods:

Overall, a lot more detail is required in relation to the study population and data collection.

  1. Provide information as to why this survey was used and describe the cohort/population chosen to which the survey was implemented on

Thanks for your comments, we have added the detailed information in this section.

  1. Please state what Hebei, Guizhou and Shandong are i.e., are these cities? Regions?

Thanks for your comments, we have added the detailed information in this section.

  1. Is this sample representative of Chinese women of the same age? If so, provide evidence that this is a representative sample

 Thank you for your comments. We have made a supplementary explanation of the sample involved schemes selected by the population representative in the paper.

  1. Out of the 2295 included, how many were not included? What total pool/sample/cohort have these women been drawn from? Please include a participant flow diagram to outline how you reached this number.

Thank you for your valuable advice to make our paper more rigorous and perfect. We have added the inclusion exclusion flow chart as Figure 1. All 2,295 people were subjects in the study.

  1. What does ‘promptly replaced’ mean and by whom and how were participants replaced? This should be incorporated into a flow diagram.

 Thank you for your review. promptly replaced means that the subjects sampled cannot participate in the survey, and we will replace them with subjects with similar ages and other aspects in the same area. We have added the instructions in the flowchart.

  1. Please provide the following details regarding study population:
  2. How were participants recruited and by whom?

Thank you for your review. Experts from the CDC in China did a sampling design. Two townships (streets) were selected from each district and county, and two village (neighborhood) committees were selected from each township (street). In each village (neighborhood) committee, 15 lactating women aged 18-50 were selected, and all of them were within 2 years postpartum. After subjects were selected, they were first asked to be a member of the survey and signed the informed consent

  1. Who randomly selected the districts?

Thank you for your review. Experts from the CDC in China did a sampling design.  China CDC staff randomly selected the districts according to demographic proportion and economic distribution.

  1. Who screened them for eligibility?

Thank you for your constructive comments. All the staff involved in the intended sample survey have received unified training and assessment from China CDC to ensure that they work in accordance with unified standards.

  1. Please report clearly how participants were to know they had dyslipidaemia? Were they screened prior to inclusion or were they supposed to know this beforehand somehow?

Thank you for your constructive comments. The diagnostic criteria for dyslipidemia refer to the "Guidelines for the Prevention and Treatment of Dyslipids in Chinese Adults (Revised 2016)"[22], that is, hypercholesterolemia: total cholesterol (TC) ≥ 6.20 mmol/L; hypertriglyceridemia: triglyceride (TG) ≥2.30 mmol/L; high-LDL-cholesterolemia: LDL-C (low-density lipoprotein cholesterol) ≥ 4.10 mmol/L; low HDL-cholesterolemia: HDL-C (high-density lipoprotein cholesterol) < 1.00 mmol/L, any of which was called dyslipidemia. Since the population in this study was lactating women, there were no individuals in the sample who were previously known to have dyslipidemia.

  1. Please provide the following details regarding data collection:
  2. It is unclear where all the data collection took place. Was this at a research facility? Out-patient clinic settings? Please provide this information.

We gathered subjects in township health centers, village clinics and community health service centers for investigation.

  1. Why were other biochemical parameters such as vitamins and CRP measured? This should be justified or explained somewhere.

Since the data source of this study is the nutrition survey, it is necessary to comprehensively monitor the relevant indicators of the nutritional status of residents. We put these indicators in Table 1 to state the basic characteristics of the survey respondents.

  1. Was weight, height, blood pressure collected serially? If so, please report

Thank you very much for your comments. This study was not a cohort study but a cross-sectional survey.

  1. Provide details on the specific questionnaires used (e.g., for physical activity, diet) and if they were validated where relevant

Thank you very much for your comments. We have added relevant content in the paper according to your comments.

  1. Report how many times did participants visit the facility for data collection

Thank you very much for your comments. We have added relevant content in the paper according to your comments. Thank you very much for your comments. We have added relevant content in the paper according to your comments.

  1. Where the blood was collected

Blood samples were collected from the median cubital vein of all subjects in township health centers, village clinics and community health service centers.

Statistics

  1. Please be specific re what ‘basic’ characteristics is referring too in lines 122

Thank you very much for your comments. We have added relevant content in the paper according to your comments. Basic characteristics included age, sex, body mass index (BMI), blood pressure, physical activity level, smoking status, drinking status, parity number, history of gestational hypertension, history of gestational diabetes, time of postpartum (age of breastfeeding child), etc.

  1. Please outline in the methods section the significance and relevance of the BMI quartiles you have chosen to use in lines 126-127

Thank you very much for your comments. We have added relevant content in the paper according to your comments.

  1. Please refer to presenting means, SD or SEM etc – as you present results in this manner but haven’t stated this in the statistical methods section

Thank you very much for your comments. We have added relevant content in the paper according to your comments.

Results

  1. Line 145, remove capital L on lactating

Thank you very much for your thoughtful and constructive suggestions. We have revised the spelling mistake as you suggested in the revised manuscript.

  1. Correct %’s and age to just 1 decimal place

Thank you very much for your thoughtful and constructive suggestions. We have revised the mistakes as you suggested in the revised manuscript.

  1. Line 152 should say ‘BMI quartiles’, line 154 correct to ‘with increasing BMI, levels of …’

Thank you very much for your thoughtful and constructive suggestions. We have revised the mistakes as you suggested in the revised manuscript.

  1. Please ensure all abbreviations have been defined in first instance

Thank you very much for your thoughtful and constructive suggestions. We have checked all abbreviations as you suggested in the revised manuscript.

  1. Table 1, 2 3, 4 – typo for Q1 should be ‘reference’

Thank you very much for your thoughtful and constructive suggestions. We have revised the mistakes as you suggested in the revised manuscript.

  1. Include adequate description of statistical tests used in Table 1 footnote

Thank you very much for your comments. We have added relevant content in table 1 footnote according to your comments.

  1. Table 1 title – suggest different wording. How is this ‘baseline’ when there is only one timepoint for this study?

Thank you very much for your thoughtful and constructive suggestions. We revised the sentence to “Characteristics by BMI quartile in Lactating Women”.

  1. Table 3 & 4, present data to 2 decimal places

Thank you very much for your thoughtful and constructive suggestions. We have revised the decimal digits in the revised manuscript.

  1. Table 3 – It would be clinically relevant to also understand the OR for women who had multiple types of dyslipidaemias to see if their OR was higher in those with higher BMI. Can you provide this information?

Thank you very much for your comments. Your point of view is worth thinking about. However, since the purpose of our study is to analyze the influence of BMI on dyslipidemia, due to the limited space of this paper, we politely believe that the OR determination of dyslipidemia on BMI could not be included. Perhaps we can study the part you mentioned in the future in an appropriate study.

Discussion

Please check the whole manuscript, and particularly Introduction and Discussion for correct English and grammar. Some punctuations like full stops are also missing.

  1. Please reword line 226, not correct grammar

Thank you very much for your thoughtful and constructive suggestions. We have revised the mistake in the revised manuscript.

  1. Need full stop in line 227.

Thank you very much for your thoughtful and constructive suggestions. We have revised the mistake in the revised manuscript.

  1. Please be consistent with in-text referencing. In the introduction you use [ ] but in the Discussion you use ( ).

Thank you very much for your thoughtful and constructive suggestions. We have revised the mistake in the revised manuscript.

  1. Please use scientific language for e.g., ‘positive tips’ in line 235; what are you trying to say here? Should be referring to the clinical translation of these findings in practice.

Thank you very much for your comments. We have revised relevant content according to your comments.

  1. Please include some discussion around the potential mechanisms of action to which you think might explain the relationship you’ve observed between BMI and dyslipidaemia. Is there anything inherent about lactation itself that may predispose women to have a higher BMI, fasting lipids or both? These areas should be discussed in light of your findings. Sections 236-253 and 254-267 only report associations/correlations, but these should be critically discussed and related back to the physiological mechanisms that might be at play here as well as the relevance to the lactation phase of a woman’s life.

Thank you very much for your comments. We have added relevant content according to your comments.

  1. Correct spelling for ‘implied’ in line 251

Thank you very much for your thoughtful and constructive suggestions. We have revised the mistake in the revised manuscript.

  1. Line 269 – please refer to the BMI classification range, not just the cut points in lines 281 to 282. i.e., are you saying that in China, the BMI for overweight is over 24kg/m2?

Thank you for your professional comments. This part describes the definition of overweight in China and in the world, so that we can compare with the cut-off value in our research for reference later.

  1. It should be explained in the methods or discussion why BMI of 40 or higher were excluded.

Thank you very much for your comments. We have added relevant content according to your comments.

Thank you for your professional comments. In this study, The number of individuals of 40 is very small, and if included, it will become an extreme value to interfere with the stability of the research results.

  1. Lines 290-292 are vague. Please describe in detail what ‘unified work manuals’ are and what types of ‘unified training’ was required, what is ‘provincial quality control groups’ and what are the ‘links’ you’re referring too? These can be added to the methods section.

Thank you for your professional and meticulous advice. We have added relevant content to the methodology section as you suggested.

  1. The lack of dietary data is a significant limitation of this study and should be considered. Diet plays a heavy role in cardiometabolic risk factors such as BMI and dyslipidaemia, and moreover, dietary recommendations are different during the lactation life stage; which could influence both BMI and dyslipidaemia independently and interchangeably. It should be explained why dietary data was not collected and the potential impact of dietary intake should be discussed in the discussion.

Thank you for your professional and meticulous comments. Your comments are very in-depth and worth thinking about. At first, we considered that BMI reflected the long-term nutritional status of subjects, and there may be multicollinearity between dietary factors and BMI. Meanwhile, cross-sectional dietary factors in the analysis may not reflect the long-term nutritional status of the subjects. We have added a note to the paper according to your suggestion.

  1. Include how these findings translate into practice, and how now knowing this cut off point of 24.85kg/m2could inform care and management of lactating women by health care professionals.

Thank you very much for your comments. We have added relevant content according to your comments.

Round 2

Reviewer 2 Report

Please see attached word doc with reviewer responses/further comments in 'blue text' underneath each respective area.

Authors, please highlight and quote line numbers for ANY and ALL changes you make. It was only done for some, thus making it frustrating to locate where or if the relevant changes were made.

Author Response

Thank you for your careful and patient review. We have revised these question.

ntroduction

  1. Overall,very short and limited in detail in terms of providing a strong rationale for this proj

Thank you for your guiding suggestion. We have modified and added the corresponding content in the introduction according to your opinion.

The introduction still lacks critical evaluation of the literature. Please attend to this comment and highlight every and any change you make so the reviewer can easily see what you have changed.

Thank you very much for your patient, meticulous and professional review. We have made the changes and highlighted them in the original text, and we recommend that you turn on the revision mode so that all the details of the changes can be displayed.

  1. TheEnglish tense used in the introduction doesn’t fi Past tense such as ‘was’ is used everywhere when referring to findings about linking BMI and dyslipidaemia. For e.g., the first line shouldn’t be past tense i.e., change to ‘body mass index IS a common measure…’ same in lines 40 & 41, 48 and others. Sentences appear unfinished e.g., line 48

Thank you very much for your thoughtful and constructive suggestions.  We have revised the grammar as you suggested in the revised manuscript.

Please highlight everywhere where you have corrected for grammar. I can also still see areas where the grammar/English is still not correct. E.g., lines 61 should be ‘There are clear STUDIES showing that …’ Please highlight every and any change you make so the reviewer can easily see what you have changed.

Thank you very much for your patient, meticulous and professional review. We have made the changes and highlighted them in the original text, and we recommend that you turn on the revision mode so that all the details of the changes can be displayed.

  1. Literaturepresented in the first paragraph of introduction requires more elaboratio For e.g., line 46 should outline what that previous literature is in relation too and how is this relevant to your project?

Thanks for your enlightening guidance, we have arranged and modified the article paragraphs according to your suggestions.

Again, still not detailed enough. In line 48, what type of population is this? Did they have a condition, adults, ethnicity? Please highlight every and any change you make so the reviewer can easily see what you have changed.

Thank you very much for your patient, meticulous and professional review. We have made the changes and highlighted them in the original text, and we recommend that you turn on the revision mode so that all the details of the changes can be displayed.

  1. Beclear what you mean by ‘frequency of dyslipidaemia gradually increased with increasing BMI’ in line 46-7 and provide a little more information on the study details e., is this a meta-analysis? RCT? Humans? Men and women? Conditioned? Etc This level of detail should be in the entire introduction.

Thank you for your review. We have added more details of the literature according to your suggestion.

Again, this comment has not been attended too – those lines read the same as before – please highlight any changes you have made so the reviewer can easily see what you have changed.

Thank you very much for your patient, meticulous and professional review. We have made the changes and highlighted them in the original text, and we recommend that you turn on the revision mode so that all the details of the changes can be displayed.

  1. Ithasn’t been convincingly justified why this phase of life (lactating) is critical to determine the optimal BMI range/upper limit with respect to dyslipidaemi Suggest adding

more information around the potential physiological implications (or even hypotheses) of excessive fat tissue (and discuss this in relation to body fat tissue / central obesity) to both the mother and baby; present what is the physiological link between BMI and dyslipidaemia; and what about other related cardiometabolic markers?

Thanks for your guidance, we have added corresponding literature and descriptions in the original text according to your comments.

Line 65 is not scientific – ‘it is easy to cause’ should be re-phrased. Please highlight every and any change you make so the reviewer can easily see what you have changed.

Thank you very much for your patient, meticulous and professional review. We have made the changes and highlighted them in the original text, and we recommend that you turn on the revision mode so that all the details of the changes can be displayed.

Methods:

  1. Isthis sample representative of Chinese women of the same age? If so, provide evidence that this is a representative sample

Thank you for your comments. We have made a supplementary explanation of the sample involved schemes selected by the population representative in the paper.

Have no idea where this is, please reference line numbers and highlight so it can be checked.

Thank you very much for your patient, meticulous and professional review. We have made the changes and highlighted them in the original text, and we recommend that you turn on the revision mode so that all the details of the changes can be displayed. In this part, we have added the sample size calculation and the details of the sampling part to illustrate the representative consideration of sampling for Chinese lactating women.

  1. Outof the 2295 included, how many were not included? What total pool/sample/cohort have these women been drawn from? Please include a participant flow diagram to outline how you reached this numb

Thank you for your valuable advice to make our paper more rigorous and perfect. We have

added the inclusion exclusion flow chart as Figure 1. All 2,295 people were subjects in the study.

  • Flowchart doesn’t make se
  • Our flowchart details how we screened available data from a preexisting nutrition surveillance population of lactating women who met our requirements. The intuitive impression may be different from your previous study design, we are using existing monitoring data for data processing and analysis. The nature of this study is similar to the use of data from the US NHANES program.
  • Startingand final numbers do not add u
  • Our flowchart details how we screened available data from a preexisting nutrition surveillance population of lactating women who met our requirements. The intuitive impression may be different from your previous study design, we are using existing monitoring data for data processing and analysis. The nature of this study is similar to the use of data from the US NHANES program.
  •  
  • Numberof subjects ‘promptly replaced’ should be provided
  • Promptly replaced occurs in the sampling stage, and each survey point is required to not exceed 5% of the amount. No specific value is counted, and we have made a supplementary explanation in the article
  • howmany excluded for each reason should be provide
  • Our flowchart details how we screened available data from a preexisting nutrition surveillance population of lactating women who met our requirements. The intuitive impression may be different from your previous study design, we are using existing monitoring data for data processing and analysis. The nature of this study is similar to the use of data from the US NHANES program.
  •  
  • DefineCNHS abbreviation
  • This abbreviation is not referred to in the text

What does pro mpt ly r epl ac ed? me an and by who m an d ho w wer e par t ic ip ant s r eplac e d?

This should be incorporated into a flow diagram.

Thank you for your review. promptly replaced means that the subjects sampled cannot participate in the survey, and we will replace them with subjects with similar ages and other aspects in the same area. We have added the instructions in the flowchart.

Details of how subjects were approached, contacted, screened etc should be provided. It is all still unclear how these women came to know about the study.

 Thank you for your review.The subjects were approached, contacted, screened etc through local CDC departments and community health service centers.

  1. Pleaseprovide the following details regarding study population:
  • Howwere participants recruited and by whom?
  1. Whoscreened them for eligibility?

Thank you for your constructive comments. All the staff involved in the intended sample survey have received unified training and assessment from China CDC to ensure that they work in accordance with unified standards.

This doesn’t answer my question. Who screened participants/subjects for eligibility to be included (or not eligible) for participation in this study/survey? This should be added to the manuscript - Please highlight every and any change you make so the reviewer can easily see what you have changed.

 Local CDC staff screened participants/subjects for eligibility to be included (or not eligible) for participation in this study.

  1. Pleasereport clearly how participants were to know they had dyslipidaemia? Were they screened prior to inclusion or were they supposed to know this beforehand somehow?

Thank you for your constructive comments. The diagnostic criteria for dyslipidemia refer to the "Guidelines  for  the  Prevention  and  Treatment  of  Dyslipids  in  Chinese  Adults  (Revised 2016)"[22],   that   is,   hypercholesterolemia:   total   cholesterol   (TC)    ≥   6.20   mmol/L;

hypertriglyceridemia: triglyceride (TG)  ≥2.30 mmol/L; high-LDL-cholesterolemia: LDL-C (low- density  lipoprotein  cholesterol)  ≥  4.10  mmol/L;  low  HDL-cholesterolemia:  HDL-C  (high- density lipoprotein cholesterol) < 1.00 mmol/L, any of which was called dyslipidemia. Since the

population in this study was lactating women, there were no individuals in the sample who were previously known to have dyslipidemia.

Again, not answering my question. How did you or the study staff determine that the women did (or didn’t) have dyslipidaemia? Lactating women can in theory, have dyslipidaemia.

First of all, none of the pregnant women in our study were taking lipid-lowering drugs, and we determined dyslipidemia by measuring blood lipid levels in these subjects during the study. Regular physical examination for lactating women is not common in China.

  1. Pleaseprovide the following details regarding data collection:
  • Itis unclear where all the data collection took  Was this at a research facility? Out- patient clinic settings? Please provide this information.

We gathered subjects in township health centers, village clinics and community health service centers for investigation.

Ok – so if it was multi-site then please report in the manuscript how you maintained standardized procedures across all measures.

All questionnaires and physical measurements were trained by China CDC, and blood parameters were measured by China CDC laboratories.

  1. Wasweight, height, blood pressure collected serially? If so, please report

Thank you very much for your comments. This study was not a cohort study but a cross- sectional survey.

In lines 146-151 you report that these measures were collected by CDC staff. Then please outline how many times weight, height, BP etc were collected.

 We have added relevant content in the paper according to your comments.

  1. Providedetails on the specific questionnaires used (e.g., for physical activity, diet) and if they were validated where relevant

Thank you very much for your comments. We have added relevant content in the paper according to your comments.

You have provided definitions for the physical labors (lines 159-163) but you have not provided details on the specific questionnaires used (e.g., for physical activity, diet) and if they were validated where relevant.

The investigator's judgment of physical activity is based on definitions for the physical labors. There is not a plethora of other questionnaires to elaborate on.

  1. Wherethe blood was collected

Blood samples were collected from the median cubital vein of all subjects in township health centers, village clinics and community health service centers.

Add what training the blood collector had

The blood collectors in this study were not temporarily trained but were medical staff who had been engaged in blood collection for years.

Statistics

.   Please refer to presenting means, SD or SEM etc – as you present results in this manner but haven’t stated this in the statistical methods section

Thank you very much for your comments. We have added relevant content in the paper according to your comments.

Cannot locate this change. Should be a sentence in the stat section – please highlight

Thank you very much for your comments. We have highlighted this part.

Results

  1. Correct%’s and age to just 1 decimal place

Thank you very much for your thoughtful and constructive suggestions. We have revised the mistakes as you suggested in the revised manuscript.

This hasn’t been done - %’s are still to 2 dec places – e.g line 210-211

Sorry for our mistake. We have revised the mistakes as you suggested in the revised manuscript.

  1. Line152 should say ‘BMI quartiles’, line 154 correct to ‘with increasing BMI, levels of …’

Thank you very much for your thoughtful and constructive suggestions. We have revised the mistakes as you suggested in the revised manuscript.

Still not done!! In Line 216, correct it to ‘with increasing BMI, levels of …’ and please highlight

the change!

  1. Pleaseensure all abbreviations have been defined in first instance

Thank you very much for your thoughtful and constructive suggestions. We have checked all abbreviations as you suggested in the revised manuscript.

Please highlight all abbreviations you have changes since the initial manuscript version.

Thank you very much for your patient, meticulous and professional review. We have made the changes and highlighted them in the original text, and we recommend that you turn on the revision mode so that all the details of the changes can be displayed.

  1. Table3 – It would be clinically relevant to also understand the OR for women who had multiple types of dyslipidaemias to see if their OR was higher in those with higher B Can you provide this information?

Thank you very much for your comments. Your point of view is worth thinking about. However, since the purpose of our study is to analyze the influence of BMI on dyslipidemia, due to the limited space of this paper, we politely believe that the OR determination of dyslipidemia on BMI could not be included. Perhaps we can study the part you mentioned in the future in an appropriate study.

Ok. Please acknowledge this in the discussion or closing remarks as a recommendation for future studies/research.

We have  acknowledge this in the discussion  as a recommendation for future studies/research.

Discussion

Please check the whole manuscript, and particularly Introduction and Discussion for correct English and grammar. Some punctuations like full stops are also missing.

  1. Pleasereword line 226, not correct grammar

Thank you very much for your thoughtful and constructive suggestions. We have revised the mistake in the revised manuscript.

No you haven’t. Reword (now line 292) to ‘Overweight and obesity in lactating women are key public health issues that deserve attention.’

We have reword this sentence and hightlighted it.

  1. Needfull stop in line 227.

Thank you very much for your thoughtful and constructive suggestions. We have revised the mistake in the revised manuscript.

No you haven’t. Please add a full-stop after ‘obesity [22]’ before ‘It’ now in line 293

Thank you. We have added a full-stop.

  1. Pleaseinclude some discussion arou
  2. ndthe potential mechanisms of action to which you think might explain the relationship you’ve observed between BMI and dyslipidaemi Is there anything inherent about lactation itself that may predispose women to have a higher BMI, fasting lipids or both? These areas should be discussed in light of your findings. Sections 236- 253 and 254-267 only report associations/correlations, but these should be critically discussed and related back to the physiological mechanisms that might be at play here as                   well as the relevance to the lactation phase of a woman’s life.

Thank you very much for your comments. We have added relevant content according to your

comments.

Still suggest more discussion here. What you have added in lines 308-311 is good but elaborate on  what  specific  ‘metabolic  enzymes’ you’re  referring  too.  Is  there  anything  other  potential mechanisms at play you can add OR lifestyle-related?

Thank you for your valuable comments. In this paragraph, we would like to discuss some contents of the mechanism more, so we apply to discuss these only and hope to get your permission.

  1. Itshould be explained in the methods or discussion why BMI of 40 or higher were exclude

Thank you very much for your comments. We have added relevant content according to your comments.

Again, I can’t locate where you’ve made changes for this one. Please highlight and refer to line numbers.

Thank you for your professional comments. In this study, The number of individuals of 40 is very small, and if included, it will become an extreme value to interfere with the stability of the research results.

Don’t know what you’re referring to. I was asking you to provide a justification as to why you excluded women with a BMI of 40 kg/m2 or more – please add this to manuscript and highlight the changes.

We have added literature and highlighted it.

  1. The lack of dietary data is a significant limitation of this study and should beconsidered.

Diet plays a heavy role in cardiometabolic risk factors such as BMI and dyslipidaemia, and

moreover, dietary recommendations are different during the lactation life stage; which could

influence both BMI and dyslipidaemia independently and interchangeably. It should be

explained why dietary data was not collected and the potential impact of dietary intake should

be discussed in the discussion.

Thank you for your professional and meticulous comments. Your comments are very in-depth

and worth thinking about. At first, we considered that BMI reflected the long-term nutritional

status of subjects, and there may be multicollinearity between dietary factors and BMI.

Meanwhile, cross-sectional dietary factors in the analysis may not reflect the long-term

nutritional status of the subjects. We have added a note to the paper according to your

suggestion.

Diet and BMI would not be multicollinear. Correlating height with BMI or weight with BMI

would be multicollinear. Therefore, please remove reference to that, as I do believe that is

wrong. Dietary intake is potentially a mediating effect on the relationship between BMI and

blood lipids, and thus ought to be acknowledged. Please edit the last sentence you added to

this: ‘In addition, future research should explore the interplay between dietary intake, BMI and

risk of dyslipidemia in lactating women’.

Thank you for your guidance. We have deleted the inappropriate sentence and added the sentence you proposed and highlighted it.

Round 3

Reviewer 2 Report

Thank you for attending to the comments. Attached are some further minor edits required, they have been outlined in green text.

Author Response

Please edit line 62 to be ‘can influence cardiometabolic disease in the mother and her children. Therefore, further understanding around the health problems of nursing women is required. Based on…etc etc

Edit line 71 to ‘BMI as a convenient and potentially relevant index’.

Thank you for your guiding suggestion. We have modified these according to your suggestion.

  • I understand. But nonetheless, all of the numbers should be accounted for. At a minimum, please provide how many were excluded for each exclusionary criteria i.e., age >/= 18 years, BMI etc as per SPIRIT Guidelines.
  • Also, in the flow chart, shouldn’t BMI cut-off sign be corrected to ‘≥’ 40, nor ‘<’ 40 ? If so, please correct.

Thank you for your guiding suggestion. We have modified these according to your suggestion.

Details of how subjects were approached, contacted, screened etc should be provided. It is all still unclear how these women came to know about the study.

 Thank you for your review.The subjects were approached, contacted, screened etc through local CDC departments and community health service centers.

Please add this to the manuscript if you haven’t already.

Thank you for your guiding suggestion. We have modified these according to your suggestion.

Local CDC staff screened participants/subjects for eligibility to be included (or not eligible) for participation in this study.

Please add this to the manuscript if you haven’t already.

Thank you for your guiding suggestion. We have modified these according to your suggestion.

Please add this to the manuscript if you haven’t already and explain at what point of the recruitment process you screened and determined they dyslipidemic status by measuring their blood lipids.

Thank you for your guiding suggestion. We have modified these according to your suggestion.

Given physical activity is a key contributor to cardiometabolic health outcomes, and a validated questionnaire to assess physical activity levels was not used, please acknowledge this in the limitations section.

Thank you for your guiding suggestion. We have modified these according to your suggestion.
